# Exploring the Evolution of the Topics and Research Fields of Territorial Development from a Comprehensive Bibliometric Analysis

Claudia Jazmín Galeano-Barrera [1,*], María Eugenia Arango Ospina [2], Edgar Mauricio Mendoza García [3], Dewar Rico-Bautista [4,*] and Efrén Romero-Riaño [5]

1 Instituto de Investigación Xerira, Facultad de Ingenierías y Tecnologías, Universidad de Santander, Bucaramanga 680003, Colombia
2 Programa Colombia Científica, Centro de Estudios Ambientales, Universidad Autónoma de Manizales, Manizales 170002, Colombia; maestriadesarrolloregional@autonoma.edu.co
3 CORE—School of Management, Facultad de Ciencias Económicas, Administrativas y Contables, Universidad Autónoma de Bucaramanga, Bucaramanga 680011, Colombia; m.mendoza@unab.edu.co
4 Grupo de Investigación GRIITEM, Departamento Sistemas e Informática, Universidad Francisco de Paula Santander Ocaña, Ocaña 546552, Colombia
5 Observatorio Colombiano de Ciencia y Tecnología, Bogotá 111311, Colombia; eromero@ocyt.org.co
* Correspondence: claudia.galeano@udes.edu.co (C.J.G.-B.); dwricob@ufpso.edu.co (D.R.-B.)

**Abstract:** Countries with great deficiencies in development, research, and innovation are investing resources to advance in this aspect; meanwhile, it is necessary to advance in initiatives that promote local development, through the correct use and management of endogenous territorial capacities to achieve economic, social, and environmental development, and this is where the territorial development approach has intervened during the last decades. To obtain an understanding of the evolution of the research field on territorial development, a study of research topics and groups of research topics is implemented from subscription-based data sources (Scopus, Science direct, Ebsco, and Web of Science) and open access (Lens and Dimension platforms). Keyword co-occurrence techniques were implemented, and indicators of link strength and density-centrality of clusters were used to abstract patterns of change within the field of study. The evolution of the area, trends, and issues related to territorial development are identified in the different databases, mapping a discipline that still lacks comparative publications between research results at different scales.

**Keywords:** territorial development; topics evolution; co-word analysis; bibliometrics analysis; vosviewer; R-Bibliometrix





## 1. Introduction

Territorial development integrates the individual and collective initiatives of the actors of a territory, which seek social construction based on cooperation between the economy, technology, politics, culture, society, and environmental and geophysical particularities of a particular place [1]. Its main purpose is to identify endogenous capacities to create institutional cohesion and among actors to build networks that work together to achieve common objectives, applying the systemic approach between the different dimensions of sustainable development within the territory [2].

The theoretical bases are based on the development of the local economy and development poles, the localization theory, clusters with new industrial districts, and social construction [3,4]. From the fundamentals, issues that strongly connect with the territory and the actors and institutions are identified, such as competitiveness, technological innovation, institutional development, the demand for goods and services both internally and externally (theory of the export base), urban-rural links, and social construction from the consolidation of relationships that reaffirm different levels [5].

In the construction of decent and sustainable development established by the 2030 agenda, the territory is a major player despite the multiple debates that can be generated around this term [6]. However, the territory is conceived as a living organism, with a sense of appropriation, with individual and collective expectations, taking into account the plurality of the territory at different scales from local to global [7], integrating the bases of economic development to endogenous capacities both urban and rural.

Few countries in the world have managed to reduce rural poverty; in Latin America and other areas worldwide, rurality is synonymous of poverty. Therefore, encouraging the development of this type of zone is urgent [8]. There is a strong link between rural territorial development and cultural identity, valuing the products and services present in the territory from its rural area, and promoting economic development at different levels, despite the risk of losing the culture and tradition required by the homogenization of a globalized world [9].

Rural territorial development is defined by Schejtman & Berdegué [4] as a "process of productive and institutional transformation of a given rural space, whose purpose is to reduce rural poverty". In consequence, the territory gains protagonism since the bet is based on the use of endogenous capacities to articulate in a sustainable and competitive way the economy of the territory, based on the cohesion between actors and institutions, both local and external to the territory.

Several metaphors are constructed to understand the evolution of society in rural or urban areas and its relationship with the science. One of these high impact metaphors was proposed by Pierre de Bordieu, who states that society is divided into relatively homogeneous and autonomous fields that have their own form of capital for which the actors or "agents" fight [10]. Capital includes tangible forms such as money and goods and intangible forms such as relational capital. Like society, science can be understood as a field composed of scientific subfields.

Scientific fields are often divided into disciplines/specialties, e.g., an artistic field can be divided into types of arts, schools, etc. In the scientific field, habitus are the dispositions shared by agents (such as language, publications, collaboration practices, etc.). They shape the perception, understanding, and actions of scientific agents and the field itself.

Just as it is possible to study and understand the evolution of social groups through linguistic, social, or economic variables, it is possible to understand the relationships and changes in research fields through the analysis of linguistic terms [11] or "components" that represent concepts, methodologies, technologies, and specific techniques of an area of study. Among the most studied linguistic components are author keywords that reflect the personal point of view of scholars and keywords indexed by editors of specialized journals and databases, e.g., WoS Clarivate and Scopus.

One of the ways to understand the evolution of language within a field of study is to study the thematic evolution through the scientific visualization of networks of mature and emerging words or terms in each period [12]. Using statistical techniques of association and clustering, it is possible to identify thematic groups composed of linguistic expressions that represent problems of frequent interest among scholars. Academic understanding of scientific publication dynamics centered in territorial development research field gave a more concrete opportunities to identify and examine the relationship between topics and the emerging and declining of research fields. Several studies address this direction in research fields such as environmental [13], social media [14], and cybersecurity [15] using a similar approach.

Conversely, the evolution of a problem-oriented field of knowledge in the social sciences is influenced by the intellectual research tradition/paradigm/research program of the research community itself. On the other hand, it is also driven by developments in neighboring and related knowledge domains.

The habitus in science generates pedagogical actions that orient the lines of research towards the choice of a fraction of research objects (topics), propose a design of problems to be solved, and the criteria to be applied in the design of the problems to be solved [10].

This description has a certain closeness to the conjectures of Thomas Kuhn when he stated that during periods of normal science, there was a reigning paradigm for a long period of time until the occurrence of a revolution and the substitution of the previous paradigm [16]. A paradigm is a distinct set of concepts or thought patterns, including theories, research methods, postulates, and standards for what constitutes legitimate contributions to a field.

Several literature reviews on the evolution of the topic of territorial development research have been identified. Most of the existing reviews focus on qualitative analysis and manual content analysis [17,18] from tools such as forums, interviews [19], and analysis of public policy documents [20–22]. There is little evidence of quantitative studies supported by machine-based methods that address the analysis of the evolution of the topic of territorial development. Based on the identification of this gap, the present research is developed as an academic contribution to the discipline by mapping science.

It is important to identify the evolution of territorial development research topics to understand the interaction between the thematic groups that are linked to the discipline and the new fields and directions in which it has evolved. In addition, it is necessary to evaluate the links between rurality and the topics of territorial development. Based on this premise, the research seeks to answer the following questions:

What has been the evolution of publications and topics related to research in territorial development? Towards what new fields and directions has research in territorial development evolved? Are rurality issues related to territorial development? Bearing in mind that the greatest efforts in this area are concentrated on urban development.

The analysis will be carried out by means of co-occurrence trends (identifying the gradual development of knowledge over time and towards which new fields and new research directions research has advanced) by searching from six scientific metadata sources and content analysis of the identified references. The vosviewer and bibliometrix software are used as tools for data processing.

Various strategies are implemented to improve the results of the content analysis processes of the scientific literature. One of these strategies is called "research profiling" which is understood as a method to scan large amounts of literature to broaden the understanding of the research domains and their patterns [23,24].

To implement this approach, a general research profile is built [23], which allows identifying the countries, institutions, and authors with the greatest participation in the publication of documents. This profile summarizes the statistics extracted from each database and is presented as a case study. The combination of the qualitative approach to visualization and the quantitative orientation of the research profile generates a contribution within the research field because it allows abstracting the complexity of large amounts of information obtained from various sources, where it is possible to trace the origin of the concepts associated with territorial development, through a genealogy of knowledge approach.

## 2. Materials and Methods

In this section, we will give a brief presentation of co-word analysis to visualize the evolution of territorial development research topics and research fields. The methodological foundation of co-word analysis is the idea that the co-occurrence of keywords describes the content of the documents in a dataset. From a methodological point of view, it is therefore a matter of obtaining a simplified representation of the networks to which they give rise.

Two keywords, x and y, are said to co-occur if they are used together in the description of the same document. A simple co-occurrence count is scarce for assessing links between co-words [25]. More frequently used short expressions are systematically found in the indexed keywords of the studied documents and are favored over more complex and less frequent words [26]. A study of the possible indices led us to choose two: the total link strength estimated by the VOSviewer software algorithm version 18 created by the CWTS of the University of Leiden in the The Netherlands and Callon's density-centrality for clustering topics.

Callon's centrality measures the intensity of links between a given community and other communities. The value can be represented as a measure of the importance of a theme in the whole collection. The callon centrality was estimated based on the Equivalence index. Equivalence index ($e$) is a measure of similarities between $i$ keywords. The Equation (1) for measuring the equivalence index is presented below.

$$e_{i,j} = \frac{c^2_{i,j}}{c_i\,c_j} \tag{1}$$

$e_{i,j}$ is the measure of similarities between items in frequencies of keyword's co-occurrences.
$c_{i,j}$ is the number of publications in which two keyword $i$ and $j$ co-occur.
$c_i$ is the number of publications in which the keyword $i$ occurs.
$c_j$ is the number of publications in which the keyword $j$ occurs.
The Equation (2) for measuring Callon centrality is presented below.

$$c = 10 \times \sum e_{nm}, \tag{2}$$

c is the Callon centrality.
$e_{nm}$ is the equivalence index between keywords $n$ and $m$.
$n$ and $m$ are keywords that belong to different themes.
Callon's density measures the internal strength of the community. This value can be represented as a measure of the theme's development. The Equation (3) for measuring Callon density (d) is presented below.

$$d = 100 \times \sum \frac{e_{i,j}}{w}, \tag{3}$$

Keyword $i$ and $j$ belong to the same theme.
$w$ is the number of keywords in the theme.
$e_{i,j}$ is the equivalence index between keywords $i$ and $j$.
Based on own research results, the Vosviewer software is identified as the scientific visualization tool with the greatest scientific impact within the last 10 years in scientometric and scientific visualization. This tool enables normalization by means of the measure of strength association. It is interpreted as a measure of similarity between the units of analysis [27]. Equation (4) expresses the strength of association between nodes through the relationship between the weight of the number of links of a node $a_{i,j}$, with the expected number of links from all nodes in the network $e_{i,j}$.

$$association\ strenght(i,j) = S_{ij} = \frac{a_{i,j}}{e_{i,j}}, \tag{4}$$

$a_{i,j}$: Actual number of links between items $i$ and $j$
$e_{i,j}$: Expected number of links between items $i$ and $j$
The association strength is proportional to the ratio between the observed number of co-occurrences of objects $i$ and $j$, $a_{i,j}$ and the expected number of co-occurrences of objects $i$ and $j$, $e_{i,j}$ under the assumption that occurrences of $i$ and $j$ are statistically independent. The Equation (5) for measuring expected number of co-occurrences is presented below.

$$e_{ij} = \frac{k_i k_j}{2m}, \tag{5}$$

$k_i$: Total number of links of item $i$
$m$: Total number of links in the network
Vosviewer and R-Bibliometrix support data mining of keywords and co-word analysis. The aim of the co-word analysis is to highlight the relationships between the keywords that can be considered as the most significant. In this way, the aim is to locate the subgroups of words that are closely linked to each other and that correspond to centers of interest

or to territorial development research problems that are the subject of major efforts by researchers [28].

Among them, we must think of a procedure to isolate the subgroups of words (which are represented by nodes) that are more strongly internally linked to the nodes that are not part of the subgroup. This attribute is known as centrality.

The description of the dynamics and evolution of a field of knowledge can be inferred by identifying these subgroups of words to characterize their content and follow their trajectories, reorganization, and emergence. To simplify the analysis, we propose an additional distinction, whose objective is to select the topics and groups of topics with a strong capacity to structure the general network. In this sense, within the main network generated from the territorial development data collection, we identify clusters with high connecting power, which play an essential role in the transformation of a network [11].

The visualizations shown correspond to graphs or networks that are composed of nodes and links. In this case, a node represents a word, and a link represents the existence of a link or co-occurrence of two words in two or more documents. Within networks, clusters are differentiated by colors that indicate similarity. Clusters are sets of nodes and links. The strength of links between two words is the intensity or normalized level of links between two words relative to all word links in the network.

A cluster of topics can be defined in two different ways. First, as a point in a general network, which is characterized by its position, i.e., by the set of links connecting it to other clusters/points in the general network. This attribute is usually referred to as link heterogeneity. Secondly, as a cluster formed by words linked together, which defines a dense, coherent, and robust network. This attribute is called density. We need this dual perspective of analysis to appreciate the dynamics and evolution of territorial development. Indeed, the global network can evolve in two ways [29].

Firstly, by the reorganization of relations between thematic clusters with a stable internal composition and, secondly, by the reconstruction and redefinition of the clusters of which it is composed. On the other hand, evolution can be characterized by the identification of the emergence of new clusters (whether they emerge progressively or are the result of the merger of existing clusters) or by the disappearance of clusters (which are progressively erased or split up). These two mechanisms are only rarely independent of each other. In general, a modification of the content of clusters and their topics is observed at the same time as a redefinition of the links that unite them.

This complexity of cluster change is the essence of the research. Looking at very large clusters, it is possible to speak of specialties, fields, or research topics that are stable over time. The problem we face is to identify the evolution of territorial development, and what changes and what is transformed by means of software tools and instruments that allow us to answer the question empirically. To this end, we provide the following two notions: centrality and density. These are intended to highlight the contribution of the various clusters to the structuring of the global network. Density characterizes the strength of the links between the words that make up the cluster. The stronger these links are, the greater the number of research problems corresponding to the cluster. Centrality measures the intensity of a cluster's external links with other clusters. The more numerous and stronger these external links are, the more this cluster designates a set of research problems considered to be decisive by the scientific or academic community. This means that the cluster in question is an obligatory passage point [11].

Within this study, an adapted visualization and content analysis methodology was built from [30–32], which includes five phases: (i) preparation, (ii) data analysis and visualization; (iii) deploy of the display; (iv) interpretation of scientific maps; and (v) comparison of references between scientific and institutional sources. The visualizations are constructed with VOSviewer and R-Bibliometrix software version 4.0 created by Aria M & Cuccurullo C. The data preparation includes the profiling of the data sets in the formats .ris and .csv.

The network visualizations are done through the option of overlay maps "overlay visualization" in VOSviewer. Overlay maps reflect individual attributes, nodes, or units of

analysis. These attributes reveal the scientific novelty when viewing the "average year of publication" [33]. The analysis and visualization of data include the selection as the unit of analysis of the indexed keywords (index keyword or keyword plus) and "all keywords" and the type of analysis of the co-occurrence links. Other maps are supported by R-Bibliometrix.

*Methodological Description*

Figure 1 describes the methodological approach and the respective stages in terms of the databases used, as well as the processing of the information where the data are extracted from the general profile of the research: participating countries, organizations, and authors. Subsequently, the data are standardized, creating thematic groups to be processed in the software for data processing in Bibliometrics (Vosviewer) and where the co-occurrence links between the nodes of territory and rurality are explored, to improve the understanding of the evolution of the discipline.

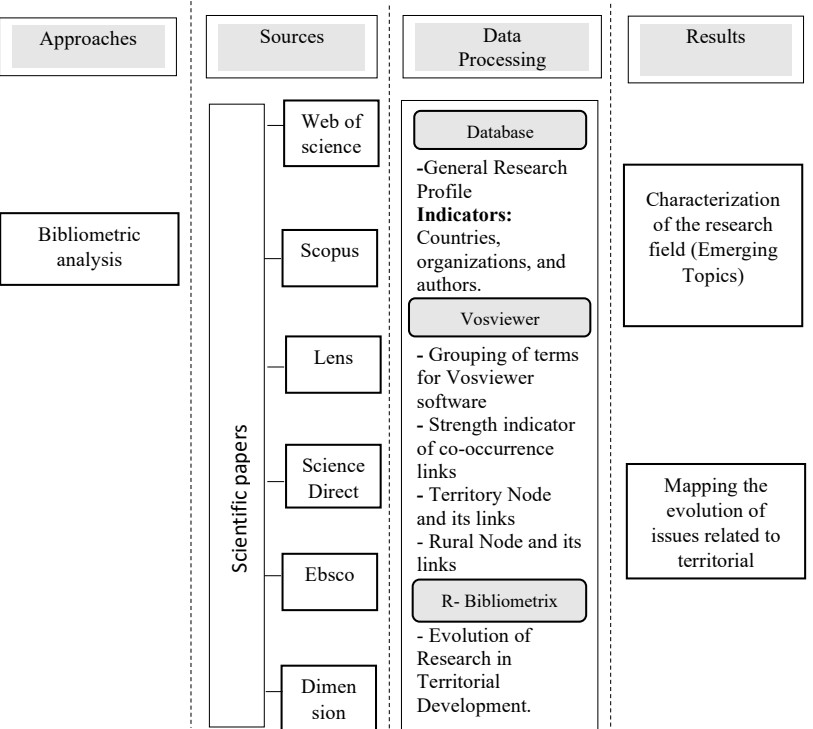

**Figure 1.** Methodological proposal based on bibliographic analysis and content analysis for the development of the research.

To obtain a broad overview of the discipline, fields, and topics derived from the publications, 6 databases have been chosen, both open access and subscription based. We have included those databases that have presented high impact and scientific relevance (open access are Dimension and Lens, while subscription databases are Scopus, Web of Science-WOS, Science Direct, and Ebsco); the difference between databases will also allow comparing results and trends between papers. Considering that only Scopus and Wos databases allow the extraction of indexed words, it was decided to use the authors' keywords in all databases as the unit of analysis for the construction of networks and visualization of topics related to the discipline.

## 3. Results

The search term "Territorial Development" is used in the different databases, hoping to abstract as many documents as possible that contain the term in their title, keywords, and/or abstracts, to identify more broadly the research topics that are linked to territorial development and to examine how rurality emerges with respect to the development of

research linked to the central theme. The dataset to be analyzed contains 20 years of publication information from 2000 to 2019. The data are recorded in Table 1, evidencing an increasing trend in discipline-related research in all the databases analyzed.

**Table 1.** Comparative of the volume of publications in relation to the search expression: "Territorial Development" according to databases.

| # Number of Publications per Database | | | | | | |
|---|---|---|---|---|---|---|
| Year | Dimensions | Lens | Scopus | Science Direct | Ebsco | WOS | Total |
| 2000 | 8 | 15 | 4 | 4 | 38 | 0 | 69 |
| 2001 | 6 | 22 | 7 | 8 | 61 | 2 | 106 |
| 2002 | 6 | 22 | 17 | 9 | 53 | 4 | 111 |
| 2003 | 8 | 30 | 11 | 9 | 64 | 3 | 125 |
| 2004 | 9 | 42 | 8 | 7 | 85 | 4 | 155 |
| 2005 | 12 | 56 | 18 | 6 | 81 | 8 | 181 |
| 2006 | 13 | 84 | 18 | 15 | 94 | 5 | 229 |
| 2007 | 27 | 88 | 26 | 14 | 115 | 12 | 282 |
| 2008 | 33 | 121 | 38 | 26 | 129 | 21 | 368 |
| 2009 | 47 | 165 | 53 | 43 | 151 | 19 | 478 |
| 2010 | 50 | 190 | 44 | 30 | 146 | 26 | 486 |
| 2011 | 43 | 211 | 55 | 26 | 207 | 24 | 566 |
| 2012 | 49 | 263 | 82 | 32 | 218 | 20 | 664 |
| 2013 | 65 | 268 | 81 | 53 | 253 | 21 | 741 |
| 2014 | 94 | 410 | 94 | 71 | 260 | 24 | 953 |
| 2015 | 123 | 423 | 118 | 103 | 255 | 118 | 1140 |
| 2016 | 125 | 425 | 120 | 129 | 266 | 109 | 1174 |
| 2017 | 133 | 337 | 142 | 100 | 303 | 114 | 1129 |
| 2018 | 195 | 410 | 171 | 114 | 261 | 139 | 1290 |
| 2019 | 267 | 327 | 133 | 119 | 182 | 101 | 1129 |
| **Total** | **1313** | **3909** | **1240** | **918** | **3222** | **774** | **11,376** |

Figure 2 shows the growth trend of all databases, a situation that shows a greater interest in generating and publishing research associated with territorial development and generating intellectual turning points at the level of this field of research [34].

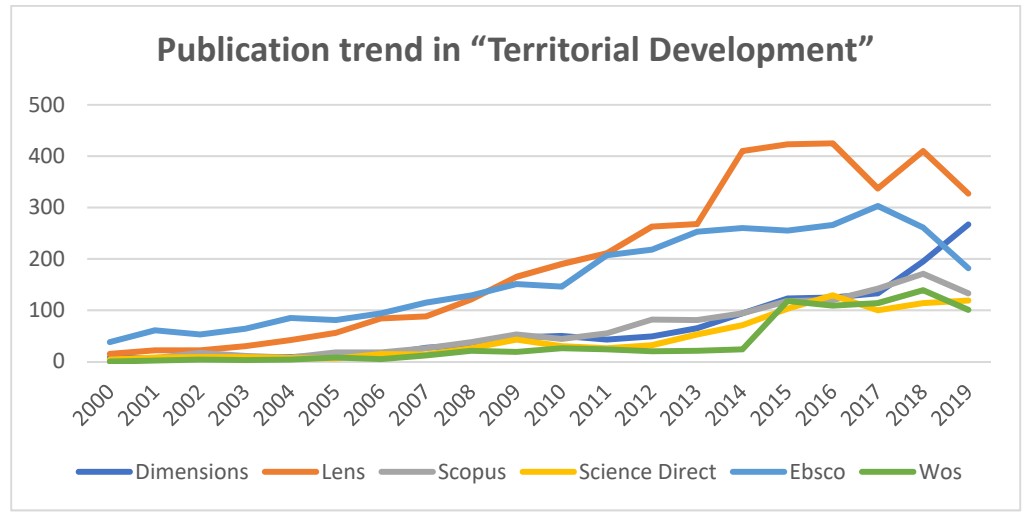

**Figure 2.** Publication trend in Territorial Development.

Among the results, it is worth noting that production increased the most in 2014, especially in the Lens database, which doubled its publication for that year with respect to the previous year.

Regarding the percentage of participation, Lens ranks first with 34.4%, followed by Ebsco with 28.3%, Dimension 11.5%, Scopus 10.9%, Science direct 8.1%, and WOS 6.8%, identifying a distribution of documents with a trend of documents with a little heterogeneity between the number of documents from open access sources (46%) vs. paid subscription access (54%).

Table 2 presents the general research profile on the production of scientific articles in the field of research on territorial development. It includes the comparison of metadata from three databases: Scopus, Lens, and WoS Clarivate. The reason for showing the information from these databases and omitting that of the other three is that WoS, Scopus, and Lens allow direct extraction of information on the geographical origin of the documents, institutions, and authors with the largest publications. The Ebsco, Science Direct, and Dimensions databases do not allow the extraction of all the data mentioned, so they are excluded from the comparison.

**Table 2.** General Research Profile.

| | DATABASE | | | | | |
|---|---|---|---|---|---|---|
| **ITEM** | **SCOPUS** | | **LENS** | | **WOS** | |
| | **COUNTRY** | **#Doc** | **COUNTRY** | **#Doc** | **COUNTRY** | **#Doc** |
| Countries | France | 193 | Italy | 260 | Spain | 135 |
| | Spain | 187 | Brazil | 132 | France | 80 |
| | Russia | 150 | UK | 130 | Italy | 68 |
| | Italy | 131 | Spain | 126 | Russia | 67 |
| | Brazil | 92 | Russia | 88 | Brazil | 64 |
| | UK | 70 | EEUU | 82 | England | 43 |
| | Portugal | 42 | France | 79 | Colombia | 29 |
| | Colombia | 37 | Colombia | 52 | Portugal | 25 |
| | Rumania | 34 | Portugal | 37 | Argentina | 33 |
| | The Netherlands | 31 | The Netherlands | 34 | Portugal | 25 |
| Organizations | Russian Academy of Sciences | 31 | University of Bologna | 94 | Russian Academy of Sciences | 30 |
| | AgroParisTech | 23 | Russian Academy of Sciences | 22 | INRA Institut National de La Recherche Agronomique | 22 |
| | INRA Institut National de La Recherche Agronomique | 23 | National University of Colombia | 19 | University of Seville | 18 |
| | CNRS Centre National de la Recherche Scientifique | 21 | Sao Paulo State University | 16 | CONICET | 17 |
| | University of Seville | 21 | Delft University of Technology | 15 | AgroParisTech | 16 |
| | University of Lisbon | 19 | Kazan Federal University | 14 | University Paris Saclay | 12 |
| | Newcastle University, UK | 17 | London School of Economics and Political Science | 13 | Universitiy of London | 12 |
| | Complutense University of Madrid | 16 | Polytechnic University of Milan | 13 | Centre National de la Recherche Scitifique CNRS | 11 |
| | Universite Grenoble Alpes | 15 | Newcastle University | 12 | National University of the South | 11 |
| | Chinese Academy of Sciences | 15 | University of Lisbon | 12 | University of Lisbon | 11 |
| Authors | Lardon, S. | 8 | Andre Torre | 18 | Medeiros, E. | 6 |
| | Medeiros, E. | 7 | A D'Orazio | 12 | Bebbington, A. | 5 |
| | Torre, A. | 7 | Andres Rodriguez-Pose | 7 | Barroso, I.C. | 4 |
| | Larrea, M. | 6 | Ameziane Ferguene | 7 | Berdegué, J.A. | 4 |
| | Bebbington, A. | 5 | Ademir Antonio Cazella | 6 | Gonzalez, R.C.L. | 4 |
| | Berdegué, J.A. | 5 | Anelise Graciele Rambo | 6 | Jeannerat, H. | 4 |
| | Jeannerat, H. | 5 | Alexandre Dubois | 5 | Torre, A. | 4 |
| | Maurel, P. | 5 | Maria Prezioso | 5 | Arbolino, R. | 3 |
| | Moulaert, F. | 5 | Ahmad Rasmi AlBattat | 4 | Barrios, J.C.D. | 3 |
| | Benneworth, P. | 4 | Alfredo Macias Vazquez | 4 | Coudel, E. | 3 |

According to the information in Table 2, the geographic origin is observed, which is defined by mapping and parsing the metadata associated with the institutional affiliation of the co-authors of the studies. Within this information, the location component of the organizations to which the authors are affiliated is analyzed. The records analyzed correspond to the countries France (193-Scopus; 79-Lens; 80-WOS), Russia (150-Scopus; 88-Lens; 67-WOS), Italy (131-Scopus; 260-Lens; 68-WOS), Brazil (92-Scopus; 132-Lens; 64-WOS), Colombia

(37-Scopus; 52-Lens; 29-WOS), and Portugal (42-Scopus; 37-Lens; 25-WOS). All of them represent countries that are very active in publications related to territorial development, occupying the top positions in all three databases. This distribution reflects the interest in territorial development research in countries with emerging economies. Other countries such as the UK (Lens-130), The Netherlands (Scopus-34), Romania, England, and Argentina also present representative publication volumes, although not simultaneously in the three databases reviewed.

The institution with the highest participation in the three databases is Russian Academy of Sciences. Other institutions with great participation are the INRA-Institut National de La Recherche Agronomique, AgroParisTech, CNRS Centre National de la Recherche Scientifique, Universiad de Sevilla, Universidade de Lisboa, and Newcastle University. Within Latin American institutions, we can find National University of Colombia (Lens-19), National Scientific and Technical Research Council of Argentina (WoS-17), and São Paulo State University, Brazil (Lens-16). Although the presence of the Russian Academy of Sciences in the three databases stands out, the volume of production of University of Bologna stands out, which is greater in a proportion close to three times.

The author with highest productivity within the three databases is André Torre (Lens-18; Scopus-7; WoS-4). Between the Scopus and WoS databases, coincidences were identified between the most relevant authors, represented in Medeiros, E. (Scopus-7; WoS-6); Bebbington, A. (Scopus and Wos-5); Jeanneret, H.; and Berdegué, J.A.

### 3.1. Construction and Visualization of Bibliometric Networks

The bibliometric maps will be displayed in the Vosviewer software from the information from the metadata of the six bibliographic data sources.

Co-occurrence analysis is used to explore changes in topics in a research field by measuring the frequency of pairs of items (i.e., substantive words or phrases) that occur throughout the body of literature in a selected field. Since the most frequent keywords provide information on the basic content of the articles, these concepts are used that perform the function of "hubs" or information centers, to articulate or group in a reasonable way topics related to rurality and the development within the research field.

For this study, these terms are grouped according to their semantic similarity within the categories of factors with the main areas of influence found in the databases (Table 3). In addition, the relationships between the axes of the study will be analyzed: territory, rurality, and territory-rurality.

**Table 3.** Grouping of terms for Vosviewer software (Thesaurus).

| Factor | Topic |
|---|---|
| Rural | agricultural development |
| | agricultural land |
| | Agriculture |
| | common agricultural policy |
| | rural development |
| | rural economy |
| | rural planning |
| | rural policy |
| Territory | territorial delimitation |
| | territorial development |
| | territorial management |
| | territorial planning |
| | territoriality |

**Table 3.** *Cont.*

| Factor | Topic |
|---|---|
| Urban | urban development<br>urban economy<br>urban growth<br>urban planning<br>urban policy<br>urban population<br>urban renewal<br>urban system<br>urban transportation |
| Region | regional development<br>regional economy<br>regional planning<br>regional policy |
| Policy | policy analysis<br>policy approach<br>policy development<br>policy implementation<br>policy making<br>policy strategy<br>politics |
| Planning | planning method<br>planning process<br>national planning |
| Economic | economic activity<br>economic analysis<br>economic and social effects<br>economic development<br>economic geography<br>economic growth<br>economic integration<br>economic theory |
| Environmental | environmental impact<br>environmental impact assessments<br>environmental management<br>environmental monitoring<br>environmental policy<br>environmental protection<br>environmental regulations |
| Land use | land management<br>land use change<br>land use planning |
| Local | local economy<br>local government<br>local participation |
| Spatial | spatial analysis<br>spatial distribution<br>spatial planning<br>spatial structure |

Next, the result of the content analysis of the similarity or relationship of terms is presented based on the "strength of links" statistic provided by the VOSviewer software. By constituting a statistic that normalizes or relativizes the importance of an "item" node within a network, its use allows abstracting a more objective view of the research field, minimizing the distortion generated by the frequent use of short or usual words, described

by Zipf's law [35]. Next, the analysis of each of the databases is summarized as a case and where Figures 3–13 correspond to keyword co-occurrence networks. The nodes of these networks are demarcated with colors indicating in yellow the emerging topics, green and blue shades topics in an intermediate range, and in purple older topics. These networks allow visualizing the general panorama of the topics addressed by the publications in each of the databases. Vosviewer estimates the association strength indicator between pairs of words; this indicator is used to build Tables 4–7.

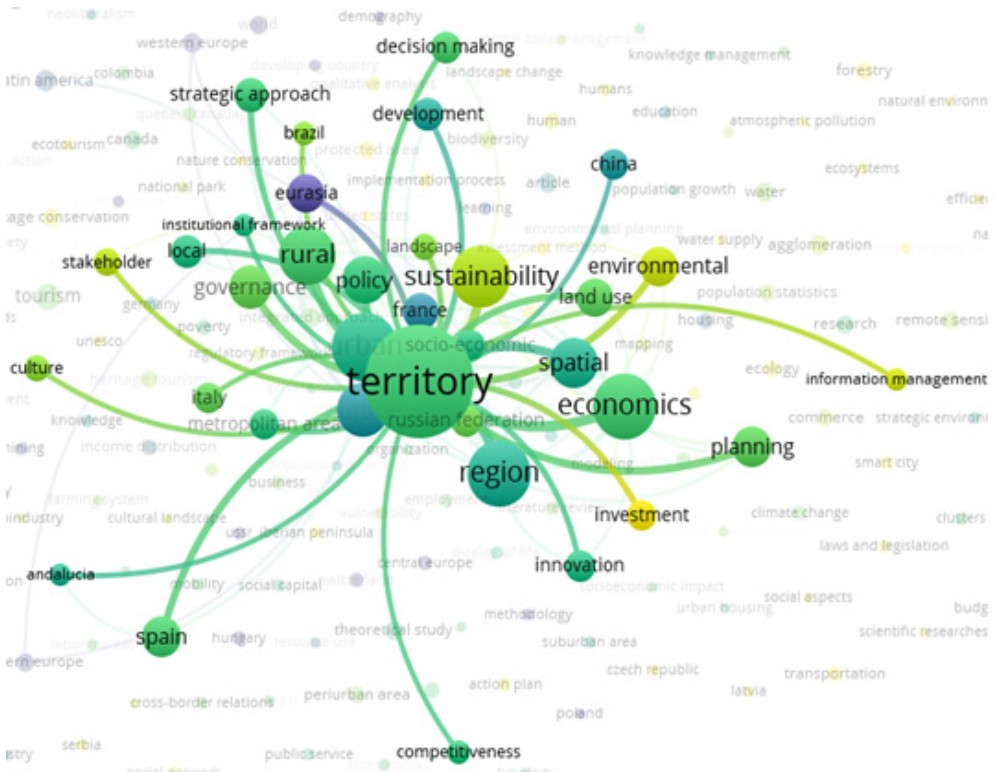

**Figure 3.** Territory Node and its links—Scopus Case. Taken from Vosviewer software.

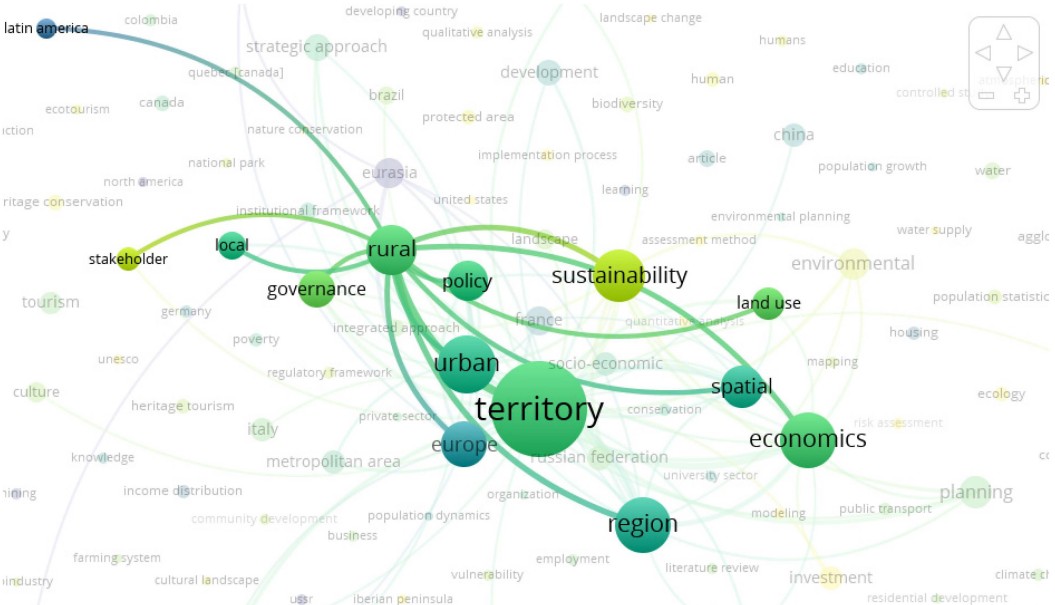

**Figure 4.** Rural Node and its links—Scopus Case. Taken from Vosviewer software.

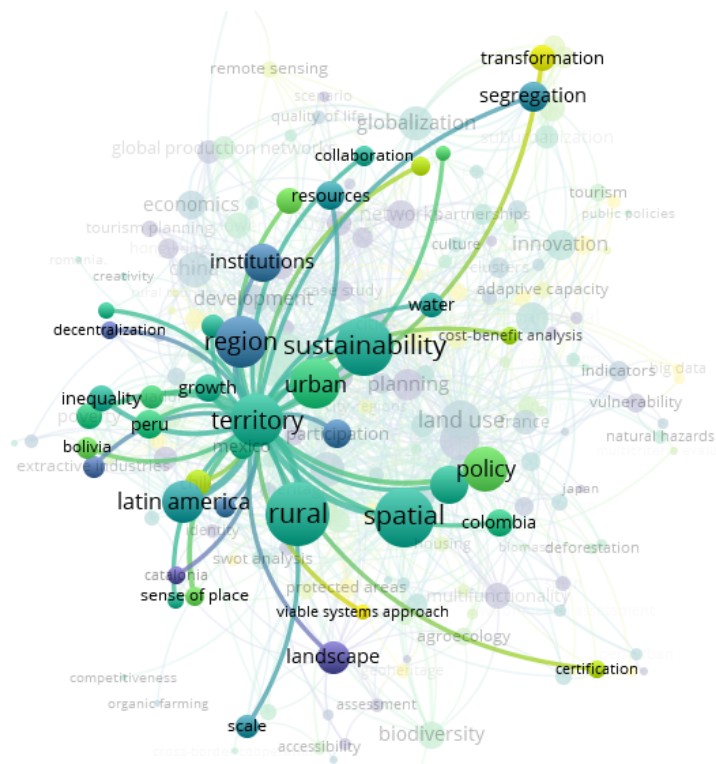

**Figure 5.** Territory Node and its links—Science Direct Case. Taken from Vosviewer software.

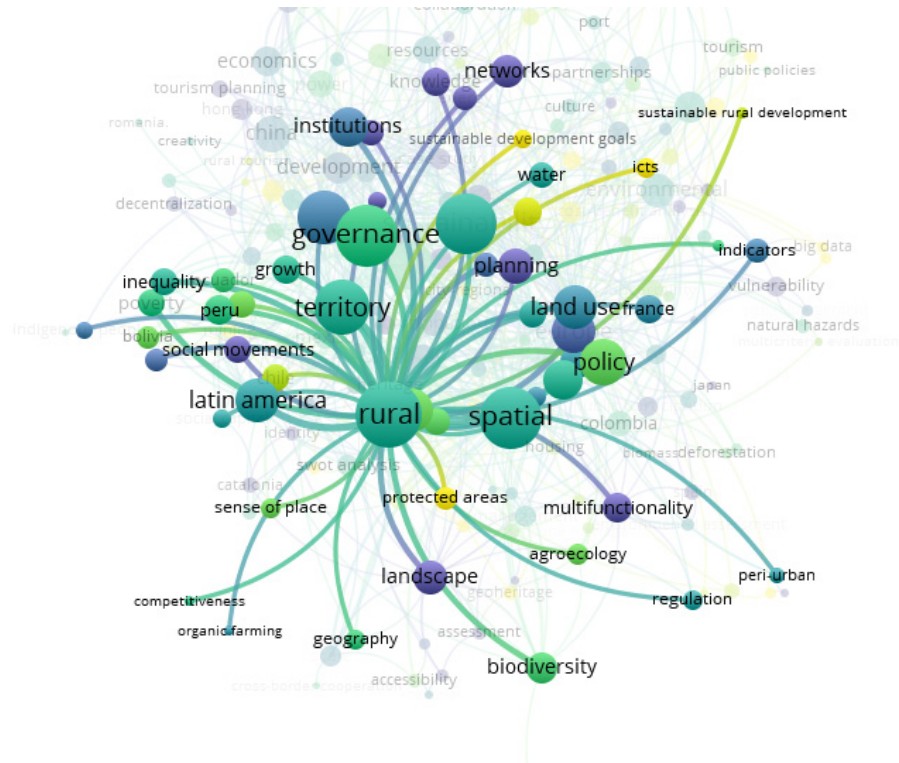

**Figure 6.** Rural node and its links—Science Direct case. Taken from Vosviewer software.

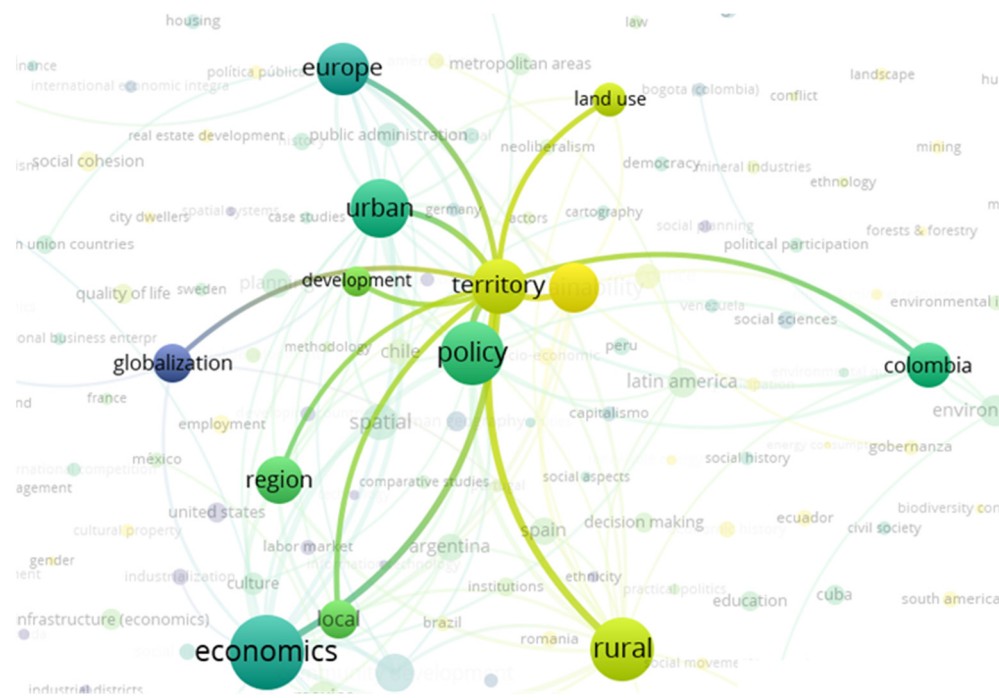

**Figure 7.** Territory Node and its links—Ebsco Case. Taken from Vosviewer software.

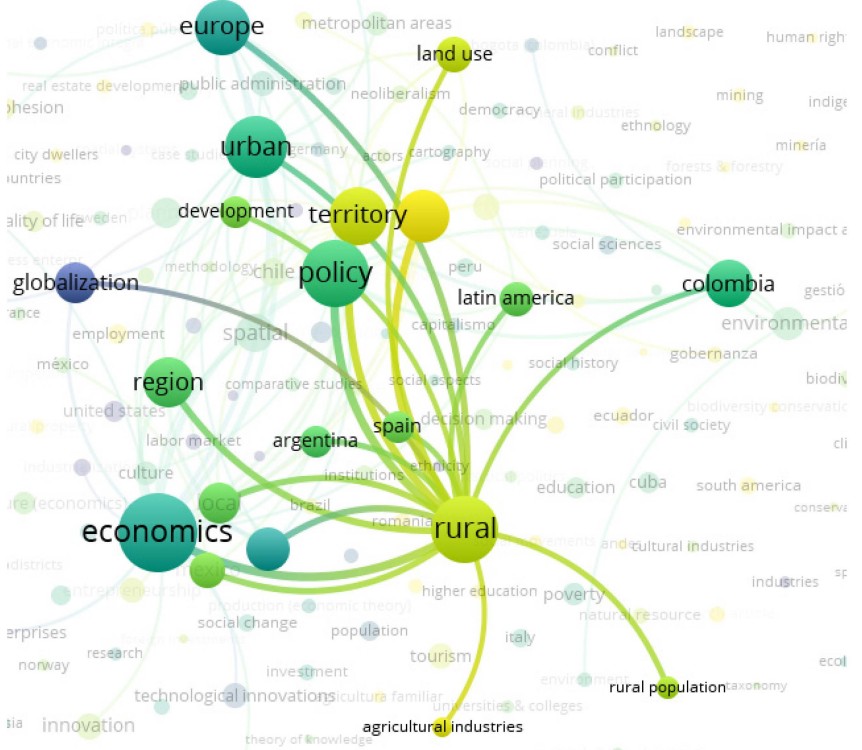

**Figure 8.** Rural node and its links—Ebsco case. Taken from Vosviewer software.

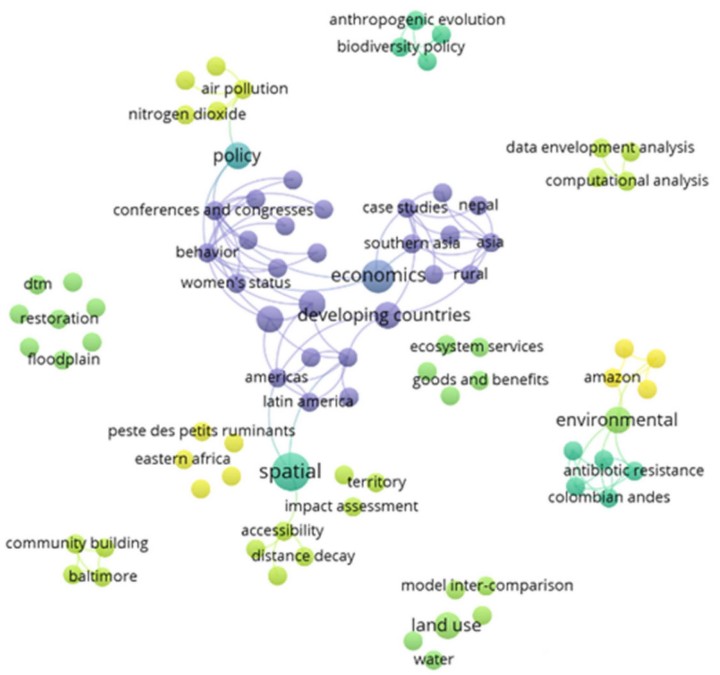

**Figure 9.** General Network—Lens Case. Taken from Vosviewer software.

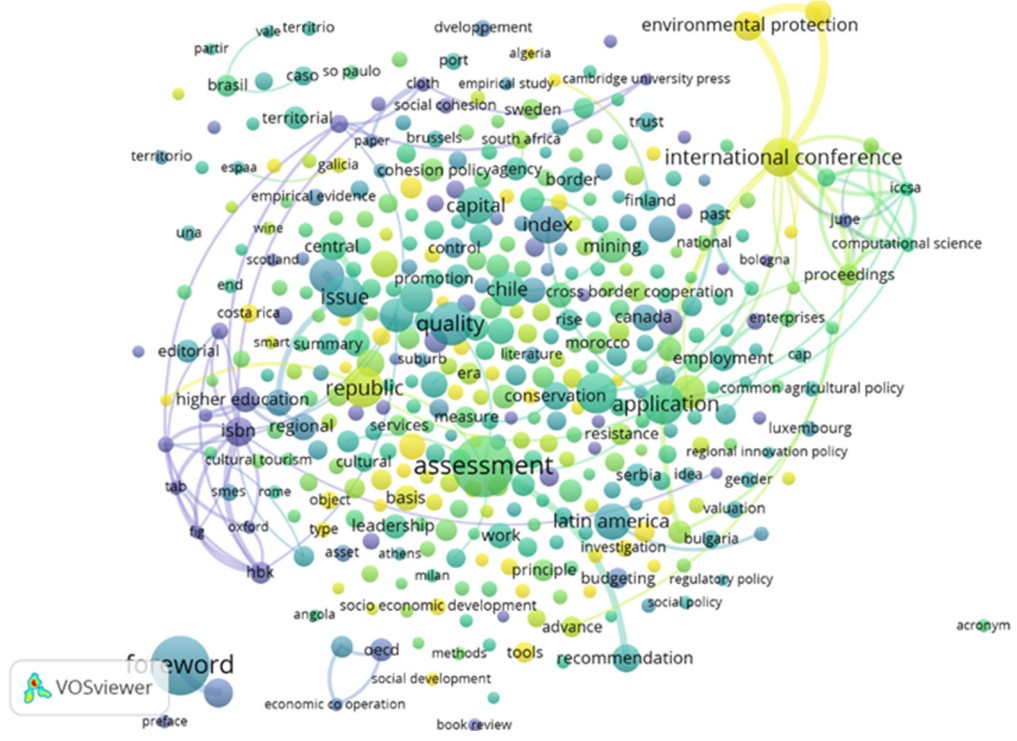

**Figure 10.** General Network—Dimensions Case. Taken from Vosviewer software.

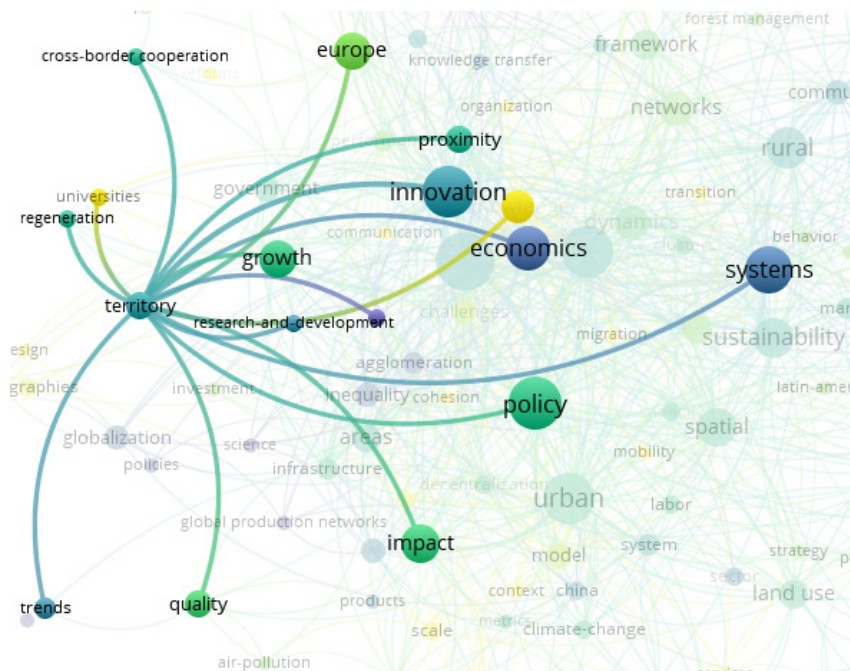

**Figure 11.** General Network—WOS Case. Taken from Vosviewer software.

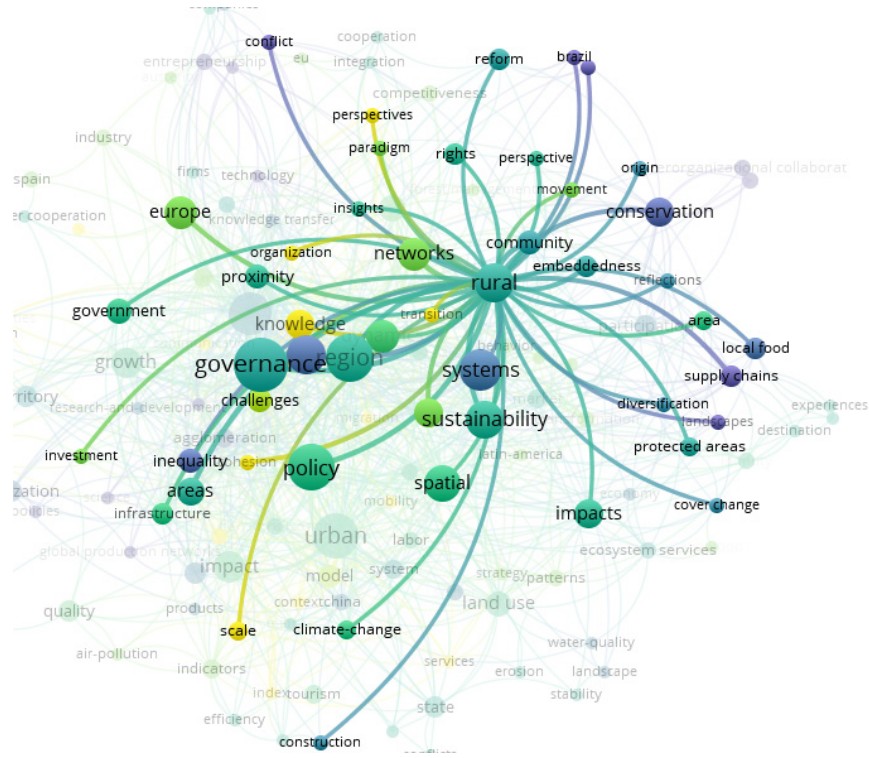

**Figure 12.** Rural node and its links—WOS case. Taken from Vosviewer software.

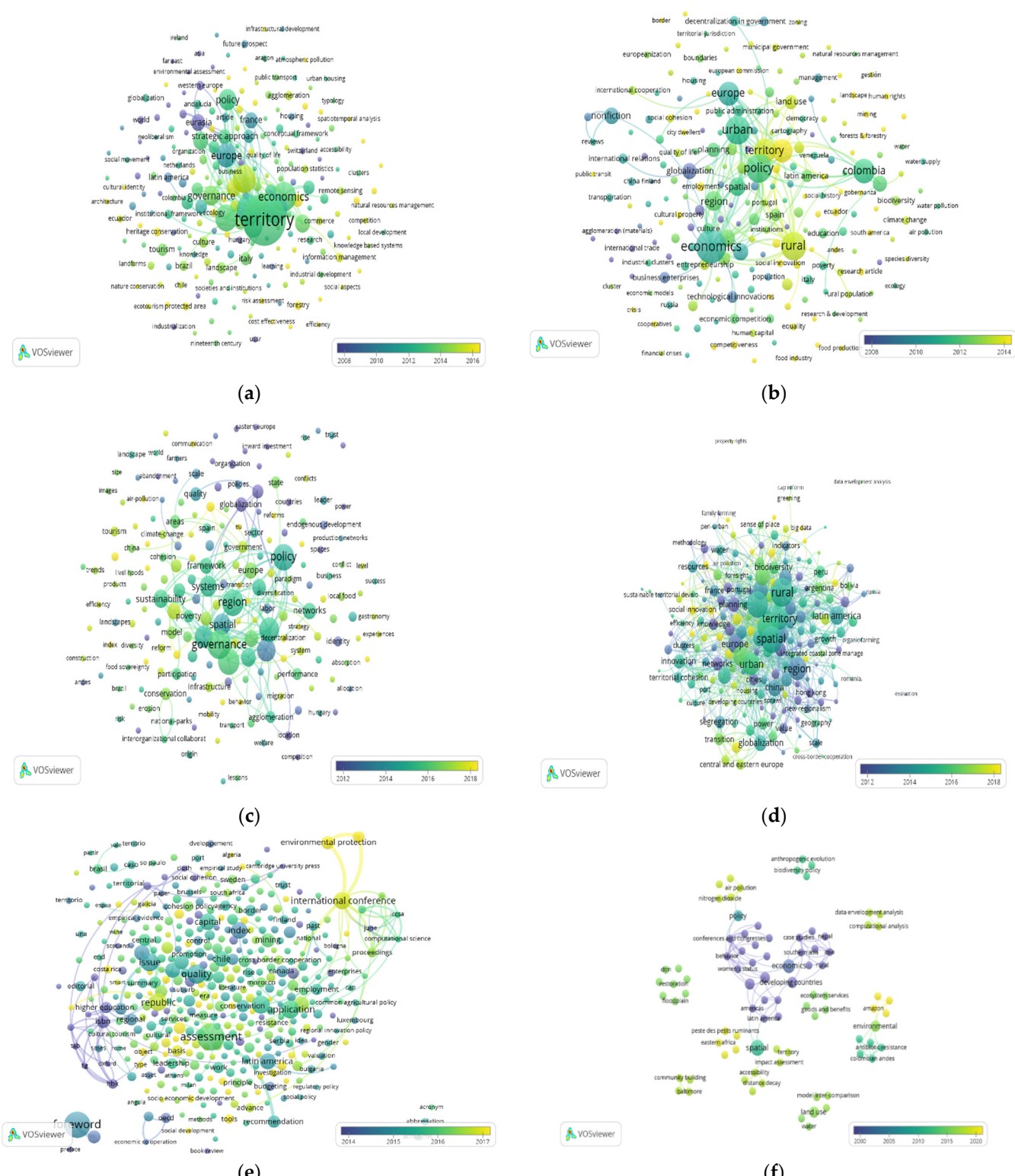

**Figure 13.** General network of each database: (**a**) Scopus; (**b**) Ebsco; (**c**) Web of Science; (**d**) Science Direct; (**e**) Dimensions; and (**f**) Lens.

**Table 4.** Cross matrix of strength of links between topical "factors" in Scopus. Own elaboration from data from the Vosviewer software.

| Factor | (1) | (2) | (3) | (4) | (5) | (6) | (7) | (8) | (9) | (10) |
|---|---|---|---|---|---|---|---|---|---|---|
| Territory (1) | - | 65 | 71 | 90 | 41 | 82 | 84 | 44 | 28 | 55 |
| Sustainability (2) | 65 | - | 20 | 23 | 14 | 27 | 30 | 0 | 0 | 12 |
| Rural (3) | 71 | 20 | - | 13 | 13 | 20 | 28 | 11 | 12 | 15 |
| Urban (4) | 90 | 23 | 13 | - | 11 | 29 | 26 | 16 | 0 | 20 |
| Governance (5) | 41 | 14 | 13 | 11 | - | 0 | 13 | 15 | 0 | 0 |
| Economics (6) | 82 | 27 | 20 | 29 | 0 | - | 49 | 12 | 10 | 20 |
| Region (7) | 84 | 30 | 28 | 26 | 13 | 49 | - | 14 | 0 | 16 |
| Policy (8) | 44 | 0 | 11 | 16 | 15 | 12 | 14 | - | 0 | 13 |
| Land Use (9) | 28 | 0 | 12 | 0 | 0 | 10 | 0 | 0 | - | 11 |
| Spatial (10) | 55 | 12 | 15 | 20 | 0 | 20 | 16 | 13 | 11 | - |

**Table 5.** Strength of the links between factors in the Science Direct case. Own elaboration from data from the Vosviewer software.

| Factor | (1) | (2) | (3) | (4) | (5) | (6) | (7) | (8) | (9) |
|---|---|---|---|---|---|---|---|---|---|
| Territory (1) | - | 2 | 5 | 2 | 0 | 2 | 4 | 2 | 0 |
| Sustaninability (2) | 2 | - | 2 | 2 | 4 | 1 | 3 | 1 | 1 |
| Rural (3) | 5 | 2 | - | 0 | 3 | 1 | 1 | 3 | 3 |
| Urban (4) | 2 | 2 | 0 | - | 0 | 1 | 6 | 2 | 0 |
| Governance (5) | 0 | 4 | 3 | 0 | - | 3 | 3 | 3 | 2 |
| Region (6) | 2 | 1 | 1 | 1 | 3 | - | 2 | 1 | 0 |
| Spatial (7) | 4 | 3 | 1 | 6 | 3 | 2 | - | 3 | 4 |
| Polity (8) | 2 | 1 | 3 | 2 | 3 | 1 | 3 | - | 1 |
| Ecosystem services (9) | 0 | 1 | 3 | 0 | 2 | 0 | 4 | 1 | - |

**Table 6.** Strength of the links between factors in the Ebsco case. Own elaboration from data from the Vosviewer software.

| Factor | (1) | (2) | (3) | (4) | (5) | (6) | (7) | (8) | (9) |
|---|---|---|---|---|---|---|---|---|---|
| Territory (1) | - | 25 | 33 | 19 | 9 | 18 | 23 | 0 | 13 |
| Economics (2) | 25 | - | 43 | 39 | 37 | 25 | 55 | 18 | 0 |
| Rural (3) | 33 | 43 | - | 16 | 23 | 30 | 39 | 0 | 9 |
| Urban (4) | 19 | 39 | 16 | - | 13 | 29 | 31 | 15 | 14 |
| Region (5) | 9 | 37 | 23 | 13 | - | 9 | 14 | 0 | 0 |
| Sustainability (6) | 18 | 25 | 30 | 29 | 9 | - | 17 | 0 | 14 |
| Policy (7) | 23 | 55 | 39 | 31 | 14 | 17 | - | 0 | 9 |
| Spatial (8) | 0 | 18 | 0 | 15 | 0 | 0 | 0 | - | 0 |
| Land Use (9) | 13 | 0 | 9 | 14 | 0 | 14 | 9 | 0 | - |

**Table 7.** Strength of the links between case factors Web of Science (Wos). Own elaboration from data from the Vosviewer software.

| Factor | (1) | (2) | (3) | (4) | (5) | (6) | (7) | (8) | (9) | (10) | (11) |
|---|---|---|---|---|---|---|---|---|---|---|---|
| Territory (1) | - | 0 | 0 | 1 | 0 | 2 | 0 | 1 | 0 | 1 | 0 |
| Region (2) | 0 | - | 2 | 4 | 2 | 4 | 2 | 0 | 4 | 3 | 1 |
| Rural (3) | 0 | 2 | - | 3 | 3 | 0 | 1 | 2 | 0 | 2 | 2 |
| Policy (4) | 1 | 4 | 3 | - | 8 | 5 | 2 | 1 | 2 | 2 | 3 |
| Governance (5) | 0 | 2 | 3 | 8 | - | 1 | 2 | 2 | 4 | 1 | 1 |
| Innovation (6) | 2 | 4 | 0 | 5 | 1 | - | 1 | 2 | 3 | 6 | 3 |
| Spatial (7) | 0 | 2 | 1 | 2 | 2 | 1 | - | 1 | 3 | 2 | 2 |
| Knowledge (8) | 1 | 0 | 2 | 1 | 2 | 2 | 1 | - | 1 | 0 | 2 |
| Urban (9) | 0 | 4 | 0 | 2 | 4 | 3 | 3 | 1 | - | 2 | 1 |
| Economics (10) | 1 | 3 | 2 | 2 | 1 | 6 | 2 | 0 | 2 | - | 1 |
| Sustainability (11) | 0 | 1 | 2 | 3 | 1 | 3 | 2 | 2 | 1 | 1 | - |

### 3.1.1. Case 1: Analysis of Co-Occurrence Relationships between "Index Keywords" Based on Information from the Scopus Database

Once the data from the Scopus database has been processed, the nodes that constitute the "core" of the territorial development research are identified and the value of the indicator of the strength of the co-occurrence links is extracted. The total strength of co-occurrence between the topics of the network is estimated at 8830. For a given word, "total strength of co-occurrence", is the total strength of the links of a word with other words of the network. The total link strength (TLS) attribute indicates the total strength of the co-ocurrence's links of a given word with other words. The TLS is calculated as the sum of the link strength values of all pairs of nodes in the network. Table 4 shows that the territory "item" registers strong links with the nodes: urban, regionalization, economy, rurality, sustainability, political space, governance, and land use. These topics reflect the duality of development between urban-rural topics because strategic concepts, such as "Agrópolis", promote the reduction of the quality of life gap between city and countryside as a requirement for the development of agricultural innovation [36], through strategies such as transfer through outreach and extension [37].

Concerning the above, there is a greater interest in associating development issues with the urban component of the territory and its dynamics, marginalizing the understanding of the rural, for which a framework of generalized theories is offered that ignore the particularities of what is rural, directly influencing the planning processes of rural territories, which reduces the effectiveness of the different development strategies implemented [38].

It is identified that the "core" of the topics within Scopus (keywords that have links with all the "items" in the table) is made up of two terms: "rural" and "territory". Although it is not surprising that the topics "territory" and "rural" register links with all the other items, they do highlight the topics "region-rural" and "urban-economics", which are linked to 90% of the topics. These thematic dyads reflect the concepts derived from thesauri used by databases and journals to relate development to urban and rural settings. However, the marginal relationship that underlies according to the findings of the exercise draws attention when establishing the rural-urban connection, represented by very weak links and a relationship that becomes invisible when you continue to think of the rural as the pantry of urban dwellers.

Disconnection is identified between the node "sustainability" with the topics of "Land use and policies", given the low number of publications that link the aforementioned factors. This same situation is identified for the topic "governance" with economic factors, land use, and geographic space. It is also possible to visualize the absence of links between land use and urban planning, governance, regionalization, and politics.

The central node is territory that is mainly linked to urbanism, regionalization, economy, rurality, sustainability, space, politics, governance, and land use. Additionally, it is related to development, decision making, strategic evaluation, institutions, planning, innovation, competitiveness, and how emerging issues in addition to the sustainability component were identified with investments, environment, and information management, as shown in Figure 3.

A strong relationship is observed between territorial development and the rural sector, with strong links with the region, governance, sustainability, local area, and spatial aspects related to analysis, distribution and spatial structure, and territorial ordering, as can be seen in Figure 4. The Latin American and European countries with the highest number of contributions stand out.

### 3.1.2. Case 2: Analysis of Co-Occurrence Relationships between "Keywords" Based on Information from the Science Direct Database

Analyzing the results processed from the information provided by the Science Direct database, new issues associated with the rural factor are identified. The issue of eco-systemic services emerges, which refers to the benefits that an ecosystem manages to contribute to society, positively impacting the health, economy, and quality of life of people.

Not preserving the ecosystem represents significant damage to human well-being, so it is included as a unit of analysis (node) within Table 5 of relationships that can be seen below, considering that the total strength of the network is 1135.

The node with the greatest strength in its links is the rural node, followed by the spatial node, governance, and sustainability.

The territory node presents its greatest link of strength with the rural node, followed by the spatial node, and maintains relationships with sustainability, urbanity, the region, and politics. The institutions appear as an important node, and the participation of countries such as Colombia, Ecuador, Bolivia, Peru, Chile, and Mexico are observed, as shown in Figure 5.

The territory is then a daily reality, a space built and appropriated historically, socially, and culturally by individuals, communities, and peoples, generating identity and cohesion of society, both at a rural and urban level. Therefore, society cannot exist without territory [39].

It is observed in the rural node an important relationship with the territory, ecosystem services, resilience, space, politics, land use, region, sustainability, and institutions mainly, as can be seen in Figure 6.

### 3.1.3. Case 3: Analysis of Relationships between "Keywords" Based on Information from the EBSCO Database

The Ebsco database supplies most of its publications in Spanish, so it was necessary to translate words to proceed to associate them in factors that correspond to the nodes, as was done in the exercise in Table 3, as well as to process a significant amount of data that was searched as follows: "territorial development".

With this set of data, it is observed that the node with the greatest strength and occurrence is "economics", indicating that territorial development has a strong connection with the economic factor within the published research, in addition to its connection with most nodes of Table 6 that connects with entrepreneurship, business company, technological innovations, and regionally connects with Latin America and Mexico. Next, the strength between links can be observed considering that the total strength of the network is 8452.

The territory node has its greatest strength link with the "rural" node, evidencing a strong dynamic between these two research areas; the "economics" and "policy" nodes are also significantly linked.

The territory is linked to local, regional, and global aspects that consider development, sustainability, rurality, and urbanity. Among the geographic areas that these themes interact in this dataset, Europe and Colombia are found, as shown in Figure 7.

Regarding the rural node, its main connection is with economic, political, regional, and territorial issues, associating areas such as Spain and Latin America in which Colombia and Argentina participate, as shown in Figure 8.

### 3.1.4. Case 4: Analysis of Relationships between "Keywords" Based on Information from the LENS Database

Analyzing the data from the LENS database, it is possible to observe weak and isolated connections of the main analysis nodes in this work; in this data set, the main nodes are: "developing countries", "economics", "urban", "policy", "spatial", and "environmental". However, it is not necessary to build a link strength table since the nodes mentioned above are disconnected.

In the node referring to "Developing Countries", there is associated the case study methodology, with participation of areas related to Asia, America, and the Caribbean.

Although in the previous cases the Territory is one of the central nodes, in this case, it appears as an emerging issue disconnected from the network. Figure 9 shows the map generated, indicating with yellow color the most recent issues within the processed data set.

As emerging issues, computational analysis is identified with "data envelopment analysis", with works on air pollution, bioaccumulation, accessibility, and territory mainly.

### 3.1.5. Case 5: Analysis of Relationships between "Keywords" Based on Information from the Dimensions Database

In the case of the Dimensions database, the database has limitations when trying to analyze emerging issues, since the available metadata only allow establishing relationships between authors, institutions, and countries and lack the keywords; this is an advantage because it yields a good volume of data.

The software has another option to create maps from the text data and proceeds to review the network with the data from the titles of the Dimensions documents, but the information is limited for this investigation. Figure 10 shows the map generated from the information of the titles of the documents.

### 3.1.6. Case 6: Analysis of Relationships between "Keywords Plus" Based on Information from the Web of Science—WOS Database

The data from the WOS database allowed finding new terms associated with the search objective; the territory node is related to politics and economic factors of interest, additionally with components of innovation, proximity, research and development, systems, and knowledge (Figure 11).

The configuration of the network relationships makes it possible to identify that governance and politics represent the strongest nodes and are related to each other (see Table 7). In the central position are governance with the economy, region, innovation, knowledge, politics, proximity, urban planning, rurality, systems, sustainability, space, and land use mainly.

The second node with the greatest strength is innovation, which is mainly related to knowledge, proximity, governance, economy, dynamics, region, networks, system, sustainability, policies, urban planning, territory, and clusters.

The data show a close relationship between rurality and governance; relationships emerge with new nodes associated with terms such as strategies, networks, knowledge, and comprehensiveness, and the rural node continues to be linked to nodes of interest in this study such as the economic factor and sustainability, as seen in Figure 12.

### 3.2. Comparison between Database Results

By making a comparison between the data related to territorial development in six databases, it is possible to identify factors associated with the subject such as sustainability, rurality, urbanism, governance, economy, region, policies, land use, space, ecosystem services, innovation, and knowledge.

Some trends in patterns between databases are identified regarding the processing of information through the Vosviewer software for the identification of topics, from which the following association is presented.

The databases that presented the greatest strength in their links were Scopus and Ebsco, which range from 8452 to 8830; they also presented robust nodes with a high number of occurrences and a high relationship between territory and rurality.

The Web of Science and Science direct databases presented much fewer strong links, ranging between 1135 and 1157; additionally, they generated less robust nodes with less occurrence, with the emergence of important nodes such as systems, proximity, institutions, collaboration, and transformation.

The similarity is found between the Dimensions and Lens databases, which despite being open access databases, with high volumes of information, had fewer connections, and their metadata is limited to being processed by the software. Figure 13 shows the general map of each database.

The overlay visualizations in Figure 13 are generated as a result of the use of the distribution, clustering, and normalization techniques of the VOSviewer software. Each node represents a "keyword or topic", and the size of each node is proportional to the frequency of co-occurrence of the corresponding keywords. The color of the nodes represents the attribute of the average publication date of the articles where each topic is referred to in the keywords.

To know both the oldest and most recent publication points, the automatic function of Vos Viewer is used to estimate the ranges of the average publication dates of the documents. The algorithm estimates the volume of documents concentrated in both old and recent dates and generates a colored grid, so this grid varies between databases given the difference in the concentration of documents in each of them, as shown in Figure 13.

The purple nodes indicate the topics whose average dates are at the lower limit of the range presented in the lower right corner of each graph. Based on this condition, a maturity or decline character is inferred for the referred terms within the database. In contrast, nodes in yellow have average publication dates close to the upper limit of the range. Supported in this characteristic, these terms are conferred an emergent nature.

Not surprisingly, the nodes of the term "territory" report the largest sizes within the Scopus, Ebsco, and Science Direct databases, since the standardized search string "territorial development" contains the term. In this sense, the identification of the relevant nature of the topics governance-region (WoS), evaluation-quality (Dimensions), and spatial (Lens) is revealing because it reflects specific thematic lines for the development of research in this field.

Based on the determination of the average publication dates, a decline in territorial development research is identified within traditional contexts such as Eastern and Western Europe and Canada and the emergence of interest in development studies in emerging economy countries such as Ecuador (Scopus), Costa Rica (Dimensions), Bolivia (WoS), and Amazon and East Africa (Lens). Of special interest within this study is the fact that the topics of territory and rural register an emergent character within the Ebsco database. These same topics within the Scopus and WoS databases are neutral (their average publication dates are in the middle of the interval) and are identified by the green color of the nodes. In the other databases, it was not possible to conclude around these two topics because they do not emerge within the visualized network.

The VOSviewer software uses the link strength estimator to calculate the relevance of the topic co-occurrence links. This vision complements the identification of the relevance of the topics based on their frequency of appearance within the keywords of the articles. Relevance based on the strength indicator is a normalized measure because its estimate is the product of the division between the number of links of a node concerning the expected value of the total links of the network.

Tables 4–7 summarize the research topics related to territorial development, based on the strength indicator. Since no significant links of territorial development topics are identified within Dimensions and Lens, it was not possible to perform the analysis for these databases.

The 13 topics identified in Tables 4–7 represent the "core" of the research topics within the analyzed bases. Several coincidences are identified by studying these topics. Within the Scopus, Ebsco, and Science Direct bases, there is 80% agreement around the topics: (i) territory, (ii) sustainability, (iii) rural, (iv) urban, (v) region, (vi) politics, (vii) land use, and (viii) spatial. The differences are centered on the fact that Science Direct does not identify topics on "economic" aspects and Ebsco does not identify "governance". Only within Science Direct does it refer to the concept of "ecosystem services" and only in WoS does it refer to "innovation" and "knowledge". The WoS results show the highest number of divergences since it does not include topics of rural, region, and land use.

### 3.3. Evolution of Territorial Development: A Longitudinal Analysis Maps

To understand the dynamics of the concepts on territorial development presented, a longitudinal analysis exercise is performed to identify the evolution of the topics related to territorial development over time. The exercise is performed with the consolidation of the metadata of the documents from all the databases, which were processed and unified, eliminating the duplicity of documents through the DB database manager, to be processed in the R-Bibliometrix software, which abstracts and correlates the evolution of the topics

within a specific sample. From these data, the analysis of the evolution of the terms presented in Figure 14 can be performed.

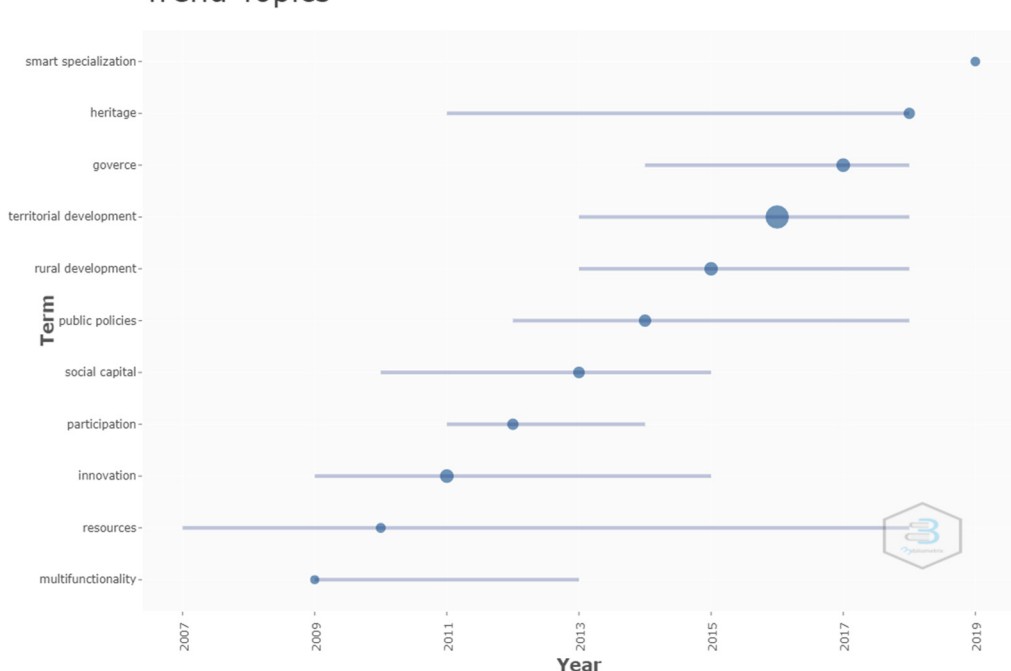

**Figure 14.** Evolution of Research in Territorial Development. Source: Own Elaboration in Software R-Bibliometrix.

The *y*-axis in Figure 14 shows the trending topics according to their frequency of occurrence. The *x*-axis represents the timeline covering each term. The size of the bubbles in the timelines reflects the frequency of occurrence of a topic. Within the results, we identified the dynamics of interest of discipline scholars in eleven specific topics. These themes are summarized in disciplines such as urbanity, geographic economics, and polycentricity. The topic of resources presents the longest timeline within the group. From 2009, the research presents a trend towards multifunctionality; innovation is incorporated into territorial development since 2009 and reaches its highest peak of publications in the year 2011. In 2012, social participation is strongly introduced in the discipline and with it, public policies for the year 2014.

It is not surprising that the topic of territorial development deploys the largest bubble in size, because the search is focused on this expression. However, it is striking how a diversification towards the topics of governance, heritage, and smart specialization emerges.

*3.4. New Fields and New Research Directions in the Discipline*

Within thematic maps, fields or disciplines are presented as clusters or bubbles that agglutinate highly similar topics. According to Callon [11], it is possible to detect changes in a discipline from three situations: (i) when the strength between the links of the topics shows significant variations over periods of time, (ii) when there is a reorganization between the topics that alters the clusters, (iii) and when new topics and clusters appear.

To capture the evolution of the topics in the 20-year period, the analysis has been segmented into two periods of 10 years each, identifying the clusters or research fields, considering the three indicators of strength of the links of co-occurrence of terms and the density and centrality of the fields of territorial development.

To obtain the results, the R-Bibliometrix tool was used to generate maps of subject clusters in four quadrants. The position of a field in the upper quadrants (*y*-axis) of the

map reflects a high degree of development or density, while the groupings towards the (*x*) axis represent the centrality or number of external links of a cluster.

The graphs themselves are an abstraction of term co-occurrence networks. These maps are generated from two measures: Callon's density and centrality. By identifying groups of words with high centrality and density within the analyzed set, the strategic themes of a research field are delimited. The position of a field in the right quadrants of the map reflects a high degree of centrality (external links). The importance of each of the quadrants is summarized as follows:

(i)     Niche topics: These are specialized topics, which present strong internal relationships, due to density, but external relationships are weak because of low centrality. Consequently, they have only a very slight impact on the field of study.

(ii)    Motor topics: These are well developed and consolidated themes within the discipline and therefore have a high degree of importance.

(iii)   Basic topics: These are important topics within the field of research but are not yet sufficiently developed.

(iv)    Emerging or disused topics: These are topics that are both new to the field of research and that have ceased to be integrated into the discipline.

In the quadrants of the strategic map in Figure 15, six bubbles representing clusters of topics are displayed. These bubbles are the representation of the disciplines or fields of study of territorial development between 2000 and 2009. Each bubble shows as a label its most frequent word. Thus, there are six fields of study labeled as: (i) territory, (ii) social capital, (iii) territorial development, (iv) rural territorial development, (v) regional development, and (vi) Europe.

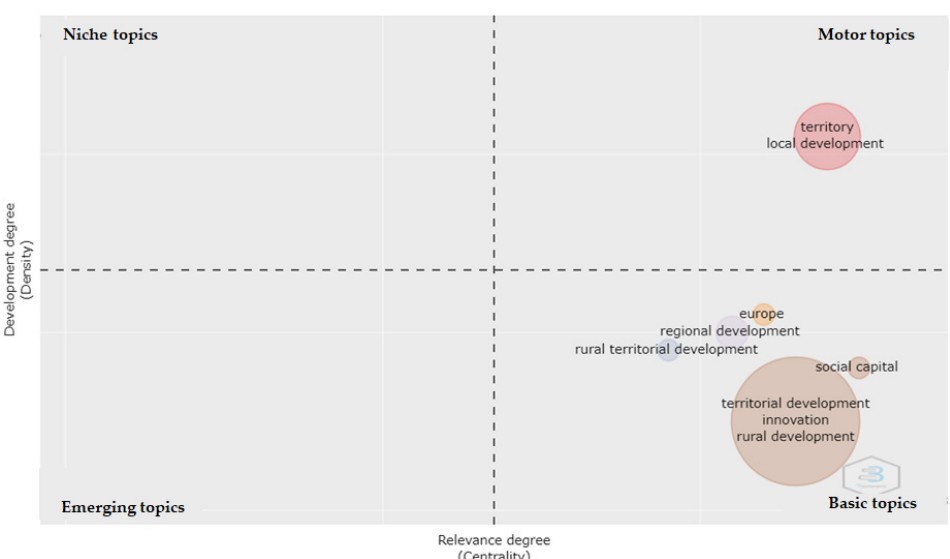

**Figure 15.** First timeline years 2000 to 2009. Source: Own Elaboration in Software R-Bibliometrix.

From the positions of the bubbles within the map, it is reasonable to suggest that the topics of territory and local development were studied together with high intensity. This is inferred from the high density or degree of development of their position on the map (density = 222.4). In turn, this field labeled as territory are those that present the greatest strength of their external links with other clusters (centrality = 2.14), served as a hub that articulated the dynamics of territorial development studies. These were the driving themes.

The topics agglomerated in the clusters located in the lower right quadrant (the core themes: regional development, rural territorial development, social capital, territorial development, innovation, and rural development) show low developmental levels in territorial development research, but high external linkages.

In the quadrants of the strategy map in Figure 16, six bubbles representing clusters of topics are displayed. These bubbles are the representation of the disciplines or fields of study of territorial development between 2010 and 2019. Each bubble shows as a label its most frequent word. Thus, there are six fields of study labeled as: (i) local development, (ii) territory, (iii) territorial development, (iv) rural development, (v) regional development, and (vi) innovation.

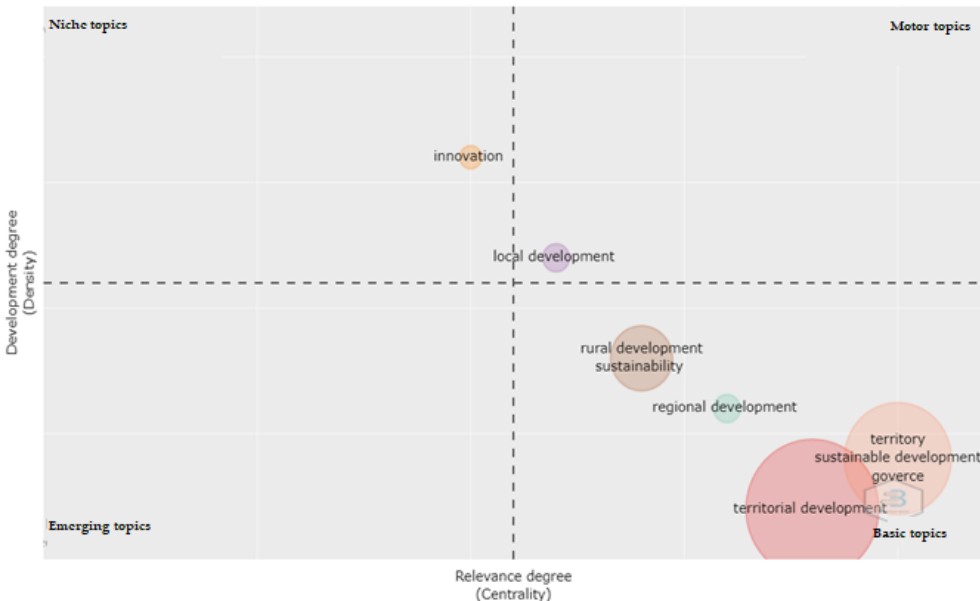

**Figure 16.** Second timeline years 2000 to 2009. Source: Own Elaboration in Software R-Bibliometrix.

Based on the comparison of the strategy maps in Figures 15 and 16, two categories of evolution of the global network of topics and research clusters in territorial development can be evidenced.

First, by the reorganization of the relationships between clusters with a stable internal composition. In the case of the territory cluster, this is bifurcated into two: territory and local development. Within the local development cluster, the driving themes of territorial research are now concentrated. Due to its emerging character, the density of internal links between the topics of this cluster is (42.35). In this sense, its degree of development or centrality is (1.94). These values are significantly lower than those of the driving themes of the previous period, partly due to the emergence of topics centered on tourism. The group of topics centered on territory went from being driving themes to basic themes. The new values of centrality and density are shown below [(density = 38.69) (intensity = 3.12)]. This shift is due to the incorporation of the topics of sustainable development and governance.

Secondly, the importance of sustainability and sustainable development issues associated with land and rural development clusters has increased. This close relationship can be explained by the effects of climate change and global warming on rural areas and their social, economic, and environmental impact.

The field of rural development remains categorized within the basic themes.

The innovation cluster emerges as the most developed set of themes according to the estimated density values (d = 51.34). The disappearance of two clusters is evident: Europe and social capital. The geographical orientation of the studies focused on Europe shows a shift towards Latin America, especially Brazil. In the case of interest in networks and social capital, a shift towards regional public policy issues is evident. In the case of the regional development cluster, it remains within the category of basic topics.

## 4. Discussion

Regarding the combination of academic subscription and free access to bibliographic sources for the analysis of research topics employing overlay maps, several challenges are highlighted.

Due to the diversity of metrics and indices and the lack of agreement or standards within the open access sources of scientific metadata, the difference in the availability of types of analysis and units of analysis between the platforms does not allow rigorous comparison of the characteristics and structure at the complete network level and at the node level (units of analysis).

Added to this situation is the challenge of incompatibility of download formats from the Dimensions (.csv) and Lens (.ris) platforms, which limit the study possibilities. In addition, it is not possible to establish equivalences between the indicators of scientific impact because their counting is carried out with criteria that differ from those of the subscription database.

Using the VOSVIEWER software, it has been possible to establish the relationships and patterns of co-occurrence of issues related to territorial development, based on a verification of the increase in publications under the specific search for "Territorial Development" in six databases where a clear positive trend is shown regarding the publication of research results.

The co-occurrence analyses made it possible to explore the evolution of research contents in territorial development through the visualization of the overlapping of networks of co-occurrence of topics using the "keyword plus" units of analysis. By examining the keywords and ranking them, the importance of 11 themes or concepts that constitute the "heart of research" within this field (rural, territory, urban, region, policy, planning, economic, environmental, land use, local, and spatial) is identified. These themes, in turn, act as "nuclei" of information and create a bridge between these consolidated themes and other emerging ones.

In terms of obtaining information, it was identified that the subscription-based academic databases (Scopus, Wos, Ebsco, and Science direct) presented more consistent results related to the extraction of metadata to be processed by the Vosviewer software, which in the case of the Open Access databases proved to be a restriction for obtaining results, despite their high volumes of information.

The focus of the research was to identify emerging issues related to territorial development and rurality, and territorial development studies topics linked to the urban and rural sectors, investing greater effort in rural development; however, it was of interest for this research to identify within the publications the connections between territory and rurality, this being a vulnerable, marginalized sector with less governmental support in countries with emerging economies [40].

It is identified that rurality presents its greatest connection with the territory in the documents published in the Scopus and Science direct databases, to a lesser extent in the documents published in Ebsco, and there is no connection with the documents published in Wos; this phenomenon can be explained by the orientation of the databases and the journals indexed in them. Despite this, it was possible to clearly determine the existence of research that has related rurality with territorial development over time and the dynamics generated between rurality and other areas of knowledge as can be seen in Figures 4, 6, 8 and 12. Among the areas that present greater strength with rurality are: territory, economy, governance, sustainability, land use, policy, and innovation.

Regarding the unit of analysis, "Territory" in general is connected with the following nodes urban, region, economy, rural, spatial, politics, governance, land use, systems, proximity, cooperation, and R + D + I. Additionally, between the emerging connecting issues include the environment, information management, sustainability, and investments; complementary topics to the previous nodes include development, decision-making, strategic evaluation, planning, innovation, competitiveness, and institutional aspects. The connections are presented in local, regional, and global aspects, spatial coordination in the sense of seeking a relationship between territories, collective action and innovation, local gover-

nance, and the company-territorial environment relationship that accounts for a dialogue between the actors, around a territorial project, which can be concretized in participatory planning that strengthens empowerment and social cohesion [41].

The findings generated from the analysis of the topics with a central position within the network of "keywords" were not novel since the topic territory presents the highest relevance based on the highest levels of the frequency indicator. On the other hand, the examination of the tables constructed from the indicator of the strength of the ties estimated by VOSviewer reflects a pattern of the relative importance of the topics centered on sustainability, rurality, urban, and politics. These results confirm the importance of the combined use of indicators within this type of study, in such a way that the contrast of the measures is guaranteed.

In addition to the connections of the rural node mentioned above, a relationship is established with ecosystem services, spatial structure, agricultural industries, integrality, networks, analysis, strategy, and weakly with the urban node.

It is identified that two of the six databases studied show a weak relationship between the rural node and the urban node (Scopus and Ebsco), while in the other databases there is no relationship at all, which demonstrates a gap between urban-rural integration in Latin America and the Caribbean, where rurality is conceived as a situation of isolation. This historical phenomenon has led to the proliferation of large and medium-sized cities with high population density, unable to meet the needs of basic goods and services and, on the contrary, to serious conditions of poverty and indigence in most of the countries of the region [41].

Analyzing the data set from the different databases, it can be observed that some of them share the same trends, producing similar results. For this particular case, the most solid results were generated by the SCOPUS databases and then by EBSCO, which showed greater strength in the links of the entire network.

These databases yielded concentrated networks, with very strong co-occurrence links; each topic is referred to multiple times within the articles, demonstrating consistency between the keywords and the disciplines in which the articles are immersed, for which they are related with highly consolidated areas of knowledge.

These two databases allow visualizing that the territory node is one of the main nodes, presenting a very strong relationship with the economy, rurality, urbanity, regionalization, sustainability, and mainly politics. Likewise, the rural node in these two databases is strongly related to the economy, sustainability, and the region as the main connections.

The second similarity in the results is observed between the Web of Science and Science Direct databases, where the territory has few connections and new themes emerge to those presented by Scopus and Ebsco, such as ecosystem services, networks, knowledge, and innovation. It is possible to identify in the results that governance retakes importance for the two cases and the network is similar in terms of its total strength indicator 1135 in Science Direct vs. 1157 in Wos, visualizing that although it presents a high number of relationships between nodes, the relationships are weak, as can be observed in Tables 5 and 7.

Although the Dimensions database presents a high number of publications in its metadata concerning the usability of the Vosviewer software, it only allows to process the titles of the documents, which generates a dispersed network with topics outside the focus of interest of territorial development. This is a case similar to the general network presented by the software with the data from the Lens database, which although it allows the processing of keywords, does not yield results that allow the visualization of units of analysis or factors related to the objective of the research.

Based on the co-citation techniques [42] of references and the estimation of indicators of network centrality of the references and authors [43], the level of influence of the documents on the concept is characterized, as well as the flows of knowledge and the evolution of the intellectual structure of the research field [34] from a genealogy approach [44]. In the case of this research through a comparative analysis of divergence of topics over time, it has been possible to identify the two major fields that have made inroads in

the discipline in recent years: sustainable development and governance. In turn, the most influential topics were determined by periods of years: Urbanity, economy, and polycentricity (2000–2008), Multifunctionality (2009), Resources (2010), Innovation (2011), Social participation (2012), Social capital (2013), Public policies (2014), Rural development (2015), Territorial development (2016), Governance (2017), Heritage (2018), and Smart specialization (2019).

The concept of territorial development involves dynamic aspects that go far beyond space. For Mazurek [41], in space nothing is neutral or trivial; space is a social production based on locations, that represents processes in permanent evolution within geographic-administrative spaces with diversity and difference as central elements of a strategy [45,46], which, as a result of a favorable environment, consolidate local economic initiatives, strengthen social capital, preserve the culture of the territory, and permanent horizontal and constructive creation from local knowledge of high added value [47,48] that allows progressing towards a balanced, lasting, and consensual growth [49].

Aguilar [41] introduces the concept of public governance as the action or process of governing, in which society and government define, on the one hand, their values, projects, priorities, agenda, futures, and course, which give meaning to society and meaning and value to associated life, and, on the other hand, the form of social organization that will allow the achievement of collective purposes based on appropriate and effective actions for the proper planning of the territory and its stakeholders; thus, governance in relation to public policies has an activating role in the territory [50].

Regarding territorial planning, it contains a strategic approach that seeks development from different dimensions, beyond the economic, as mentioned by prominent authors, including Perroux cited in [51,52]. Through his perspective, he has incorporated geopolitical, institutional and economic, and social and environmental aspects [53–55]; it is recognized as a complex and systemic process that achieves replicability, innovation, and sustainability.

Sustainability has been included in institutional documents such as the case of "Planning for sustainable territorial development in Latin America and the Caribbean" whose approach is based on equality and sustainability of territorial planning, highlighting that sustainability guarantees lasting processes over time, the result, among others, of efficient interaction with structures, actors. and institutions [56,57]. Sustainability seeks to balance the system and link the less favored areas and recognizes territorial cohesion as a determining strategy as a success factor in recent decades [58].

As an accelerated development strategy, territorial cohesion seeks to favor regions with an obvious economic lag [59] through the articulation and cooperation between sectoral policy and territorial policy, creating strong social ties based on equity and transparency with respect to diversity and territorial particularities [60].

The above is reaffirmed from the evolution of the topics presented in Figure 15 "First period of analysis" (years 2001–2009) and Figure 16 "Second period of analysis" (years 2010–2019) where the transformation of the discipline and changes in paradigms can be observed.

During the 20 years, it can be observed that regional development, rural development, and territorial development are positioned as basic topics indicating high links with other topics but still underdeveloped indicating an opportunity for research and contribution in these fields. During the second period of analysis, a disappearance of groups focused on studies on social capital and Europe is identified. The themes associated with innovation move from the basic category to the niche category, indicating their high degree of density (maturity) within the field of study. Local development themes are positioned as driving themes, replacing territorial themes, which become cross-cutting themes within the studies. Territorial themes become core themes along with sustainable development and governance. Likewise, rural development is maintained as a basic theme, but acquires greater relevance together with sustainability.

Among the paradigms associated with the transformation of the discipline is the linkage of sustainability, a topic with a high potential for development due to its broad

scope and applicability. Within the problems associated with sustainability are the effects caused by climate change and global warming within the territories and rurality, causing different effects at social, environmental, and economic levels, showing greater association with territorial development during the second period of the study.

## 5. Conclusions

In this paper, we used bibliometric data relating to 11,376 journal articles included in WoS Clarivate, Scopus, Science Direct, Ebsco Host, Dimensions and Lens, and the co-word analysis method of the bibliometric software VOSviewer to examine the development of research in territorial development. In addition, the relationships with rurality and territory were explored. The keyword analysis allowed exploring the evolution of the content of research on territorial development; the co-occurrence analysis allowed identifying the institutions that were the source of the main contributions.

The content revealed by the keywords is not surprising, although the recent interest in governance, sustainable development, strategic planning, regional economy, territorial cohesion, and tourism is indicative of the emergence of relationships with the fulfillment of sustainable development objectives contemplated in the 2030 agenda of the United Nations Organization. The comparative analysis of the most frequent and most similar topics within the field of rural development research in the six selected databases made it possible to objectively identify the scientific panorama of the research field.

Links were identified between the themes of rurality within territorial development and the themes of politics, sustainability, economics, land use, governance, biodiversity, competitiveness, agricultural industries, and globalization. The use of frequency and similarity indicators (strength of linkages), as well as network visualization techniques in overlay maps, allowed the identification of key terms within the field of study which facilitates the identification of relevant topics, as well as the characterization of their emerging or declining character within the field of research. This reveals the importance of using multiple indicators to capture the importance of the units of analysis from various perspectives.

While empirical analyses show that the volume of article production in a subject doubles every ten years, in the case of production in science of territorial development, production grew 2.3 times between 2010 and 2019. This growth rate reflects an increasing interest for scholars in territorial development issues, including different dimensions such as governance, institutions, politics, local production systems, research and development, interactions, and society.

It was established that territorial development appears in direct line with economic development and the components of competitiveness and productivity present in a territory and/or region within the framework of environmental sustainability, where the productivity of public capital in terms of Hansen [41] has a direct relationship with the level of development of the territories and the infrastructures linked to productive or economic activities explain the disparities in per capita income, with the infrastructures of the most backward regions having the greatest effects on social welfare and affecting income, a phenomenon that shows in terms of policy, the dynamizing role of public investment in the processes of regional economic development. The regional policy must fulfill its mission in terms of reducing the great regional asymmetries, prioritizing intermediate regions, where the impact is much greater so that in the medium term, they can represent new centralities that leverage the development of nearby territories with less development by becoming suppliers of specialized goods and services.

The intellectual evolution of a field of knowledge of territorial development is permeated by the traditional paradigms of its discipline; however, these are driven by related disciplines and fields of knowledge. Additionally, the evolution of the fields and the specific research topics on territorial development are affected by the global contexts in which scholars are inserted. For example, as awareness of environmental degradation has increased and climate change through global warming has become palpable, sustainability and sustainable development have emerged as research topics in territorial development.

The word co-occurrence analyses provided insight into the cumulative tradition of territorial development knowledge represented in its topics and clusters of topics understood as fields or disciplines. The evolution of the field of study of territorial development was evidenced by mapping the emergence of new thematic clusters on innovation and sustainability and topics related to tourism and governance.

Finally, the study is constrained by the choice of sources for extracting metadata from the publications. The predominance of English-language journals and articles generates an underrepresentation of topics and fields of research on territorial development in non-English-speaking countries. The papers analyzed have a weak connection with research on rural development and rurality in general.

Other limitations of the study are found in the information provided by open access sources: access is restricted and limited to titles or abstracts, excluding keywords, which was one of the units of analysis of greatest interest in the research. Likewise, not all authors report the author keywords nor do the databases report the keywords of their thesauri, generating a disparity in the selected sample.

It is suggested as future work to select only the subscription sources and unify the format of each database to obtain a single set of documents that can be cleaned and eliminate duplicate records, to process the documents in the VosViewer and R-Bibliometrix software to observe a more recent trend in the evolution of the discipline.

**Author Contributions:** Conceptualization, C.J.G.-B. and M.E.A.O.; methodology, C.J.G.-B. and E.R.-R.; validation, E.R.-R., M.E.A.O. and D.R.-B.; formal analysis, C.J.G.-B. and E.M.M.G.; investigation, C.J.G.-B. and E.M.M.G.; resources, all authors; data curation, E.R.-R.; writing—original draft preparation, all authors; writing—review and editing, all authors. All authors have read and agreed to the published version of the manuscript.

**Funding:** Scholarship 771 of 2016 for the formation of high-level human capital. Funded by the Ministry of Science and Technology of the Nation and the Government of Santander-Colombia. Scholarship 785 of 2017 for the formation of high-level human capital. Funded by the Ministry of Science and Technology of the Nation.

**Institutional Review Board Statement:** Not applicable.

**Informed Consent Statement:** Not applicable.

**Data Availability Statement:** Not applicable.

**Acknowledgments:** The authors would like to thank the Ministry of Science and Technology, Government of Santander, University of Santander, Autonomous University of Bucaramanga, Francisco de Paula Santander University, and the Colombian Observatory of Science and Technology.

**Conflicts of Interest:** The authors declare there is no conflict of interest.

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
