# Peer review of "Exploring the Evolution of the Topics and Research Fields of Territorial Development from a Comprehensive Bibliometric Analysis"

_sustainability, doi:10.3390/su14116515_

Round 1

Reviewer 1 Report

The text is very interesting. Theoretically and methodologically, it is very consistent, well organized and the results presented and discussed properly. I recommend it publication.

Author Response

Dear reviewer,

We would like to thank you for your time in the process of reviewing our article.

We are sending a new version of our paper containing the adjustments made according to the suggestions of all reviewers.

Greetings and blessings

Reviewer 2 Report

Please find my comments in the attached file

Reviewer 3 Report

The manuscript entitled “Exploring the evolution of the topics and research fields of territorial development from a Comprehensive Bibliometric Analysis” (sustainability-1681370) was aimed at exploring keyword co-occurrence through various data processing techniques, as well as at studying the evolution and trends of the scientific literature specific to territorial development by resorting to prestigious databases.

Below are my suggestions and questions regarding the submitted manuscript:

- In the abstract, please correct the following errors: "de-velopment" (line 16); "Di-mension" (line 20); "iden-tified" (line 23).

- Line 49: Why is only Latin America specifically mentioned as an example where rurality is synonymous with poverty? This is not the case of Latin America only.

- Line 226: Six scientific metadata sources and extracted data were used to carry out this bibliometric analysis, as explained in Figure 1. However, it is not clear why the authors did not merge all the articles (or at least as many as possible) from all the databases in a single file for VOSviewer processing, after eliminating the duplicates. Moreover, I suggest adding more details in Figure 1: (a) how many papers were identified per source (example: Web of Science: 774 papers).

- Are the years from Figure 13 displayed correctly: 2014 (point b); 2016 (point a); 2018 (points c and d)? It lacks consistency. Moreover. the legend is missing the Figure 13 at point e.

- "In the quadrants of the strategic map in Figure X" – please correct this error from line 745.

- In Figures 15 and 16: not all text is visible ("Niche" for example). There are no "Motor themes" previously described in lines 736-743.

- In the Conclusions section, I propose explaining the limitations of this study, as well as suggesting future research avenues.

Reviewer 4 Report

I have read this literature review carefully from beginning to end. I think it is very well done. Both the organization of the literature and the use of bibliometric methods. I have no further comments to make, at least I think it is competent.

Author Response

(The authors gave the same response as above.)

Round 2

Reviewer 2 Report

Please find the comments in the attached file 

Author Response

Please find my comments in the attached file

Reviewer 3 Report

The authors have improved the quality of the manuscript according to the suggestions, but there are still some minor issues: 

(1) The paragraph from lines 127-131 is almost identical with the one from lines 135-139 (from the PDF document).

(2) The legend (years) is missing from several figures: 5, 6, 7 (in this figure the legend is only partially present), 8, 11, 12. Kindly asking the authors for consistency throughout the manuscript.

Author Response

(The authors gave the same response as above.)

Round 3

Reviewer 2 Report

Please find the comments in the attached file

Author Response

Please find the answers in the attached file.
